

# Short Communication: Increasing vertical attenuation length of cosmogenic nuclide production on steep slopes negates topographic shielding corrections for catchment erosion rates

Roman A. DiBiase [1]

[1]Department of Geosciences, Pennsylvania State University, University Park, Pennsylvania, 16802, USA

*Correspondence to*: Roman A. DiBiase (rdibiase@psu.edu)

**Abstract.** Interpreting catchment-mean erosion rate from in situ produced cosmogenic $^{10}$Be concentration in stream sands requires calculating the catchment-mean $^{10}$Be surface production rate and effective mass attenuation length, both of which can vary locally due to topographic shielding and slope effects. The most common method for calculating topographic shielding accounts only for the effect of shielding at the surface, leading to catchment-mean corrections of up to 20% in steep landscapes, and makes the simplifying assumption that the effective mass attenuation length for a given nuclide production mechanism is spatially uniform. Here I evaluate the validity of this assumption using a simplified catchment geometry to calculate the spatial variation in surface skyline shielding, effective mass attenuation length, and the total effective shielding factor for catchments with mean slopes ranging from 0° to 80°. For flat catchments (i.e., uniform elevation of bounding ridgelines), the increase in effective attenuation length as a function of hillslope angle and skyline shielding leads to a catchment-mean total effective shielding factor of one, implying that no topographic shielding factor is needed when calculating catchment-mean vertical erosion rates. For dipping catchments (as characterized by a plane fit to the bounding ridgelines), the catchment-mean total effective shielding factor is also one, except for cases of extremely steep range-front catchments, where the shielding correction is counterintuitively greater than one. These results indicate that in most cases, topographic shielding corrections are inappropriate for calculating catchment-mean erosion rates, and only needed for steep catchments with non-uniform distribution of quartz and/or erosion rate. By accounting only for shielding of surface production, existing shielding approaches introduce a slope-dependent systematic error that could lead to spurious interpretations of relationships between topography and erosion rate.



# 1 Introduction

Measurement of in situ produced cosmogenic $^{10}$Be concentrations in stream sediments has rapidly become the primary tool for quantifying catchment-scale erosion rates over timescales of $10^3$-$10^5$ y (Brown et al., 1995; Granger et al., 1996; von Blanckenburg, 2006; Portenga and Bierman, 2011; Codilean et al., 2018). Although requiring a number of simplifying assumptions about the steadiness of erosion and sediment transport (Bierman and Steig, 1996), erosion rates determined from $^{10}$Be concentrations in stream sediments have in general shown to be robust and have yielded insight to a number of key questions in tectonic geomorphology regarding the sensitivity of erosion rates to spatiotemporal patterns of climate, tectonics, and rock strength (e.g., Safran et al., 2005; Binnie et al., 2007; Ouimet et al., 2009; DiBiase et al., 2010; Bookhagen and Strecker, 2012; Miller et al., 2013; Scherler et al., 2017).

In contrast to point measurements, where a clear framework exists for converting $^{10}$Be concentrations to either a surface exposure age or steady erosion rate (e.g., Balco et al., 2008; Marrero et al., 2016), the interpretation of $^{10}$Be concentrations in stream sediment requires accounting for the spatial variation in elevation, latitude, quartz content, and erosion rate throughout a watershed (Bierman and Steig, 1996; Granger and Riebe, 2014). Additionally, topographic shielding corrections that account for the reduction of cosmic radiation flux on sloped or skyline-shielded point samples (Dunne et al., 1999) are applied to varying degrees for determining catchment-mean production rates. These shielding corrections are either applied at the pixel level (e.g., Codilean, 2006), catchment level (e.g., Binnie et al., 2006), or not at all (e.g., Portenga and Bierman, 2011). Although typically small (<5%), topographic corrections can be as large as 20% for steep catchments (e.g., Norton and Vanacker, 2009). Because these corrections vary as a function of slope and relief, any systematic corrections can influence interpretations of relationships between topography and erosion rate.

The pixel-by-pixel skyline shielding algorithm of Codilean (2006) results in the largest topographic shielding corrections, and has gained popularity due to its ease of implementation in the software packages TopoToolbox (Schwanghart and Scherler, 2014) and CAIRN (Mudd et al., 2016), the latter of which was used to recalculate published $^{10}$Be-derived catchment erosion rates globally as part of the OCTOPUS compilation project (Codilean et al., 2018). A key simplification of the Codilean (2006) approach is that it accounts only for the skyline shielding of surface production, and not for the change in shielding with depth, which determines the sensitivity of the effective mass attenuation length for nuclide production as a function of surface slope and skyline shielding (Dunne et al., 1999; Gosse and Phillips, 2001). Because a change in the effective mass attenuation length will directly influence the inferred erosion rate of a sample (Lal, 1991), the full depth-integrated implications of topographic shielding must be accounted for when inferring catchment erosion rates from $^{10}$Be concentrations in stream sediments.

Here I model the shielding of incoming cosmic radiation flux responsible for spallogenic production at both the surface and at depth for a simple catchment geometry to evaluate as a function of catchment slope and relief the total topographic shielding



factor and the partitioning of shielding into surface skyline shielding and changes to the effective attenuation length. I then apply this framework to catchments that have a net dip (i.e., dipping plane fit to boundary ridgelines) and compare calculations of total shielding to those from typical pixel-by-pixel skyline shielding corrections.

## 2 Theory

The incoming cosmic ray intensity, $I(\theta, d)$, responsible for in situ cosmogenic nuclide production by neutron spallation can be most simply described as a function of the inclination angle above the horizon of the incident ray path, $\theta$, and the mass depth, $d$ (g cm$^{-2}$), traveled along that pathway:

$$I(\theta, d) = I_0 \sin^m \theta \, e^{-d/\lambda},  \tag{1}$$

where $I_0$ is the maximum cosmic ray intensity at the surface, $m$ is an exponent typically assumed to have a value of 2.3 (e.g.,

Nishiizumi et al., 1989), and $\lambda$ is the mass attenuation length (g cm$^{-2}$) for unidirectional incoming radiation (Dunne et al., 1999). For a horizontal surface sample ($d = 0$), the unshielded total cosmic radiation flux, $F_0$, can be described by:

$$F_0 = \int_{\varphi=0}^{2\pi} \int_{\theta=0}^{\pi/2} I_0 \sin^m \theta \cos \theta \, d\theta \, d\varphi = \frac{2\pi I_0}{m+1},  \tag{2}$$

where $\varphi$ is the azimuthal angle of incoming radiation, and the term $\cos \theta$ accounts for the convergence of the spherical coordinate system. For point samples that are either at depth ($d > 0$) or have an incomplete view of the sky due to topographic

shielding by thick ($d \gg \lambda$) objects, the total cosmic radiation flux, $F$, is modulated by a shielding factor, $S(\theta_0, d)$, such that:

$$S(\theta_0, d) = \frac{F}{F_0} = \frac{m+1}{2\pi} \int_{\varphi=0}^{2\pi} \int_{\theta=\theta_0(\varphi)}^{\pi/2} \sin^m \theta \, e^{-d(\theta,\varphi)/\lambda} \cos \theta \, d\theta \, d\varphi,  \tag{3}$$

where $\theta_0(\varphi)$ is the inclination angle above the horizon of topographic obstructions in the direction $\varphi$ and $d(\theta, \varphi)$ varies as a function of both ray path azimuth and inclination angle (Dunne et al., 1999; Gosse and Phillips, 2001).

Equation (3) has two implications for interpreting exposure ages or erosion rates from cosmogenic nuclide concentrations of samples partially shielded by skyline topography ($\theta_0(\varphi) > 0$). First, skyline shielding will reduce the surface production rate of cosmogenic nuclides by a factor of $S_0$:

$$S_0 = \frac{m+1}{2\pi} \int_{\varphi=0}^{2\pi} \int_{\theta=\theta_0(\varphi)}^{\pi/2} \sin^m \theta \cos \theta \, d\theta \, d\varphi.  \tag{4}$$

Second, due to shielding of low intensity cosmic radiation below incident angles of $\theta_0(\varphi)$, the effective mass attenuation

length, $\Lambda_{eff}$, will increase relative to the nominal mass attenuation length for describing cosmogenic nuclide production as a function of depth, $\Lambda$ (Dunne et al., 1999; Gosse and Phillips, 2001). For calculating surface exposure ages, only the reduction in surface production rate due to skyline shielding need be taken into account, and Eq. (4) is easily calculated for single points in the landscape (e.g., Balco et al., 2008). However, for determining erosion rates both the surface shielding and changing effective attenuation length must be accounted for, which requires solving Eq. (3) numerically as a function of vertical depth

below the surface, as described below.



## 3 Topographic shielding model for a simplified catchment geometry

### 3.1 Simplified catchment geometry and model setup

For stream sediment samples that require calculating cosmogenic nuclide production rates across an entire catchment, solving Eq. (3) as a function of depth is presently impractically computationally intensive. Consequently, numerical implementations

of topographic shielding calculations at the catchment scale make the simplifying assumption that $\Lambda_{eff} = \Lambda$, and thus $S = S_0$ (Codilean, 2006; Schwanghart and Scherler, 2014; Mudd et al., 2016). Here I use a simplified catchment geometry to solve Eq. (3) and calculate directly the impact of topographic shielding on interpretations of catchment erosion rates from cosmogenic nuclide concentrations in stream sediments. For simplicity, I assume that cosmogenic nuclides are produced only by neutron spallation (i.e., $\Lambda = 160$ g cm$^{-2}$) and that the erosion rate, $E$, is high enough that radioactive decay is negligible (i.e.,

$E > 0.01$ g cm$^{-2}$ yr$^{-1}$ for $^{10}$Be).

Catchment geometry is simplified as an infinitely long v-shaped valley with width $2L_h$ and uniform hillslope angle $\alpha$ (Fig. 1). Because the ridgelines have uniform elevation, there is no net dip to the catchment, the effect of which will be explored in Section 3.3. At a distance $x$ and vertical depth below the surface $z$, the shielding factor, $S(x, z)$, can be defined as:

$$S(x, z) = \frac{m+1}{2\pi} \int_{\varphi=0}^{2\pi} \int_{\theta=\theta_0(x, L_h, z, \varphi, \alpha)}^{\pi/2} \sin^m \theta \, e^{-d(z, \rho, \theta, \gamma(\alpha, \varphi))/\lambda} \cos \theta \, d\theta \, d\varphi, \tag{5}$$

where $\rho$ is rock density, here assumed to be 2.7 g cm$^{-3}$, and $\gamma$ is the apparent dip of the hillslope in the azimuthal direction $\varphi$ (Fig. 1b). The mass attenuation length for unidirectional radiation, $\lambda$, differs from the nominal mass attenuation length that describes cosmogenic nuclide production as a function of depth, $\Lambda$, due to the integration of radiation from all incident angles. Assuming $m = 2.3$, a value of $\lambda = 1.3\Lambda$ results in a close match for horizontal unshielded surfaces with exponential production

profiles typical of spallation reactions (Dunne et al., 1999; Gosse and Phillips, 2001). The inclination angle integration limit, $\theta_0$, is a function of topographic skyline shielding inclination, and can be determined geometrically (Fig. 1) as:

$$\tan \theta_0 = \begin{cases} \frac{(x \tan \alpha + z) \cos \varphi}{2L_h - x}, & 0 \leq \varphi < \frac{\pi}{2} \\ -\tan \alpha \cos \theta - \frac{z}{x} \cos \varphi, & \frac{\pi}{2} \leq \varphi \leq \pi \end{cases}. \tag{6}$$

The apparent dip, $\gamma$, can be derived from the model geometry in Fig. 1 as:

$$\tan \gamma = -\tan \alpha \cos \varphi, \tag{7}$$

and the mass distance traveled through rock by a given incident ray as:

$$d = \frac{\rho z \cos \gamma}{\sin(\theta - \gamma)}. \tag{8}$$

Equation (5) was solved numerically for a series of hillslopes over a grid of $(x/L_h = [0,1]; \rho z/\Lambda = [0,40])$ with horizontal spacing $dx = L_h/500$ and vertical spacing $dz = \Lambda/500\rho$. To characterize mean slope controls on the total shielding factor,



$S(x, z)$, the above calculation was applied to nine hillslopes with mean slope, $\alpha$, ranging from 0-80° in 10° increments. Because $L_h >> \Lambda/\rho$ for most natural landscapes, the resulting distribution of shielding factors is independent of hillslope scale.

### 3.2 Calculation of shielding parameters from model results

After applying Eq. (5) to a hillslope, it is straightforward to calculate the surface skyline shielding component, $S_0(x) = S(x, 0)$. This skyline shielding component should match the topographic shielding factor determined from the algorithm of Codilean (2006), so for comparison this parameter was calculated at each pixel in the model catchment using TopoToolbox (Schwanghart and Scherler, 2014). Two additional parameters were calculated at each slope position using Eq. (5): the effective (vertical) mass attenuation length, $\Lambda_{eff}(x)$, and the total effective shielding factor, $C_{eff}(x)$.

Although spallogenic production of cosmogenic nuclides is well-described by an exponential decrease with depth for horizontal unshielded surfaces, this is not true in general for shielded samples (Dunne et al., 1999). Thus, while not exactly equivalent, the effective mass attenuation length, $\Lambda_{eff}(x)$, can be approximated by the depth at which the shielding factor is 5% of the surface shielding (i.e., 3 e-folding lengths) such that:

$$S(x, \tfrac{3\Lambda_{eff}(x)}{\rho}) = 0.05 S(x, 0). \tag{9}$$

As a consequence of the non-exponential decrease in shielding factor with depth, it is inaccurate to use the analytical relationship between surface sample concentration, $C(x)$ (atoms g$^{-1}$), and steady-state vertical erosion rate, $E$ (g cm$^{-2}$ yr$^{-1}$), typically applied to eroding samples:

$$C(x) = \frac{S(x) P_0(x) \Lambda_{\text{eff}}(x)}{E}, \tag{10}$$

where $P_0(x)$ is the unshielded surface production rate, corrected for latitude and air pressure (Lal, 1991). Equation (10) derives from integrating the path history of a particle being exhumed vertically at a steady rate $E$ and emerging at the surface with an accumulated nuclide concentration $C(x)$:

$$C(x) = P_0(x) \int_{t_0}^{t_{surface}} S(x, z(t)) dt, \tag{11}$$

which can be parameterized in terms of depth, $z$, according to:

$$C(x) = \frac{P_0(x)}{E/\rho} \int_0^{z_0} S(x, z) dz, \tag{12}$$

where the depth of a rock parcel below the surface $z_0$ at time $t_0$ is deep enough such that there is no cosmogenic nuclide production ($z_0 = 40\Lambda/\rho$ for the calculations below) and $t_{surface} = t_0 + \rho z_0/E$ is the time it takes for a rock parcel to travel from depth $z_0$ to the surface. Because there is no analytical solution for Eq. (12), the integral needs to be solved numerically. A total effective shielding factor, $C_{eff}(x)$, acts as a correction factor to interpret local erosion rate from a sample concentration, defined by:



$$C_{eff}(x) = \frac{C_{shielded}(x)}{C_{unshielded}(x)} = \frac{\sum_{z=0}^{z=z_0} S(x,z)}{\sum_{z=0}^{z=z_0} S'(x,z)}, \tag{13}$$

where $\sum_{z=0}^{z=z_0} S'(x,z)$ is the integrated shielding depth profile for the case $\alpha = 0$ (i.e., no slope or skyline shielding), and $C_{eff}(x)$ does not depend on spatial variations in latitude or air pressure corrections. Finally, a mean effective shielding factor, $\overline{C}_{eff}$, can be defined for the whole hillslope as:

$$\overline{C}_{eff} = \frac{1}{L_h} \sum_{x=0}^{x=L_h} C_{eff}(x), \tag{14}$$

which is equivalent to the catchment-mean shielding factor for the simplified valley geometry shown in Fig. 1.

### 3.3 Approximation for dipping catchments

Although the above framework accounts for variations in catchment relief and hillslope angle, $\alpha$, in all cases there is no net dip to the entire catchment (i.e. ridgeline elevations are uniform), which is not the case for natural watersheds. To simplify the geometry of a dipping catchment, I use a similar approach as Binnie et al. (2006) to model the catchment as a plane fit through the bounding ridgelines with dip $\beta$. I focus on two end-member cases, using examples from the San Gabriel Mountains, California, USA for illustration (Fig. 2). First, for an "interior" catchment that is tributary to a larger valley within a mountain range, the catchment will have a net shielding similar to the geometry of the hillslope in Fig. 1. Consequently, the shielding geometry can be approximated by Eq. (5)-(8) with $\alpha = \beta$. For the case of an "exterior" catchment that has a net dip $\beta$ but no opposing skyline shielding, Eq. (6) becomes:

$$\tan \theta_0 = \begin{cases} 0, & 0 \leq \varphi < \frac{\pi}{2} \\ -\tan \alpha \cos \theta - \frac{z}{x} \cos \varphi, & \frac{\pi}{2} \leq \varphi \leq \pi \end{cases}. \tag{15}$$

For both examples, I compare the catchment mean shielding factor, $\overline{C}_{eff}$, to the mean surface skyline shielding factor, $\overline{S}_0$, as calculated using the commonly applied topographic shielding algorithm of Codilean (2006) in TopoToolbox (Schwanghart and Scherler, 2014).

### 4 Model results

For the catchment geometry shown in Figure 1, the local shielding factor, $S(x,z)$, decreases with increasing depth, $z$, distance downslope, $x$, and increasing slope, $\alpha$ (Fig. 3). The surface skyline shielding factor, $S_0(x)$, decreases with distance downslope, $x$, and increasing hillslope angle, $\alpha$, with the greatest shielding occurring in the valley bottoms of steep catchments (Fig. 4a). For the case $\alpha = 80°$, comparison of $S_0(x)$ with the topographic shielding algorithm of Codilean (2006) shows that the two are equivalent.

The normalized effective attenuation length, $\Lambda_{eff}/\Lambda$, decreases as a function of distance downslope and increases with increasing hillslope angle (Fig. 4b). Although for low slopes cosmogenic nuclide production is concentrated at depths of



$\rho z/\Lambda = [0, 3]$, for very steep slopes production rates at depth can be greater than those of flat landscapes despite lower surface production rates (Fig. 3). This effect emerges in part due to the increased effective attenuation length for collimated radiation in skyline-shielded samples (up to a factor of 1.3—Dunne et al., 1999; Gosse and Phillips, 2001), but mainly because on steep slopes a point at depth $z$ below the surface is receiving incident radiation from oblique pathways that can be much shorter than those overhead (Fig. 1c). Consequently, there is an additional radiation flux that increases the effective (vertical) mass attenuation length, $\Lambda_{eff}$, an effect that is most pronounced near ridgelines ($x/L_h < \sim 0.4$) where skyline shielding is minimized (Fig. 3, 4b).

The combined effect of the decrease in surface production (Fig. 4a) and the increase in effective attenuation length (Fig. 4b) leads to a pattern whereby the total effective shielding factor, $C_{eff}(x)$, is greater than one along the upper portion of hillslopes and less than one along the lower portion of hillslopes near the valley bottom (Fig. 4c). Although for steep slopes ($\alpha > 60°$) there may be considerable variation in shielding depending on slope position, the mean effective shielding parameter, $\overline{C}_{eff}$, is unity for all cases (Fig. 5a).

For the case of dipping catchments (Fig. 2), the sensitivity of the mean effective shielding parameter to catchment dip, $\beta$, depends on whether catchments are "interior" (i.e., shielded by an opposing catchment) or "exterior" (i.e., no external skyline shielding). For "interior" catchments, the shielding calculations are identical to the analysis above, and thus $\overline{C}_{eff}$ is again unity for all cases (Fig. 5a). For "exterior" catchments, the increase in effective attenuation length at steep slopes due to shorter oblique radiation pathways (Fig. 1c) is larger than the decrease in surface production due to skyline shielding, and $\overline{C}_{eff}$ is greater than one (Fig. 5b). However, for all but the most extreme catchment dips ($\beta \le 40°$), $\overline{C}_{eff}$ is effectively one (within 1%).

For the two example catchments in the San Gabriel Mountains (Fig. 2), the mean total effective shielding factor, $\overline{C}_{eff}$, is 1.00, despite steep catchment dips ($\beta = 17°$ and $32°$) and high mean surface skyline shielding, $\overline{S}_0$ ($\overline{S}_0 = 0.87$ and $0.84$ as calculated by the Codilean (2006) algorithm) (Fig. 5a).

## 5 Implications for interpreting catchment erosion rates from [10]Be concentrations in stream sediment

The above results indicate that no correction factor for topographic shielding is needed to infer catchment-mean erosion rate from [10]Be concentrations in stream sands for most cases, as long as the assumptions of spatially uniform quartz content and steady uniform erosion rate are valid. Only in the extreme case of an "exterior" catchment with mean dip $\beta > 40°$ will such corrections be necessary. Although the approach of calculating only the surface skyline shielding component of the total effective shielding factor is appropriate for calculating surface exposure ages, neglecting the slope and shielding controls on



the effective mass attenuation length leads to a systematic under-prediction of the actual erosion rate. The magnitude of this under-prediction increases with increasing catchment mean slope, as highlighted by a recent compilation of catchment erosion rates from steep catchments in the Himalaya and Eastern Tibetan Plateau (red data points, Fig. 5a).

For catchments with spatially variable quartz content or erosion rate, a spatially distributed total effective shielding factor, $C_{eff}$, must be calculated at each pixel. While calculating the surface skyline shielding component is straightforward (Codilean, 2006), solving Eq. (3) at depth for arbitrary catchment geometries is presently too computationally intensive to be practical. However, while not entirely transferable to arbitrarily rough topography, Fig. 4c suggests that for slopes less than 40°, the total effective shielding factor does not vary significantly across the hillslope. For steep catchments with spatially variable quartz

content or erosion rate, direct calculation of shielding at depth is likely needed to calculate the spatially distributed total effective shielding parameter.

The modeling approach above assumes a simplified angular distribution of cosmic radiation flux (Eq. (1)) and accounts only for cosmogenic nuclide production via spallation. In actuality, the cosmic radiation flux does not go to zero at the horizon, and

becomes increasingly collimated (higher $m$) with increasing atmospheric depth (Argento et al, 2015). Thus, the sensitivity of the effective mass attenuation length to shielding will increase with increasing elevation. However, the magnitude of changes in the effective mass attenuation length due to shielding-induced collimation is at most 30% (Dunne et al., 1999), compared to the factor of 3 or more increase due to slope effects (i.e., shorter oblique radiation pathways on very steep slopes; Fig. 1c; Fig. 4b). Similarly, the dependence of $\Lambda$ on atmospheric depth, which is typically not accounted for in catchment erosion

studies, is minor (<10% for catchment with 4 km of relief (Marrero et al., 2016)) compared to the above slope effect. Treatment of cosmogenic nuclide production by muons is less constrained than spallogenic production, but the angular distribution of production by muons is likely similar to that for spallation reactions and also sensitive to latitude and atmospheric depth (Heisinger et al., 2002a; 2002b).

Overall, the effect of topographic shielding corrections on interpreting catchment erosion rates is small compared to typical assumptions inherent to detrital cosmogenic nuclide methods. In particular, the assumption of steady lowering is likely to be increasingly inappropriate for steep landscapes characterized by stochastic mass wasting (Niemi et al., 2005; Yanites et al., 2009), an effect that requires the non-trivial calculation of spatially distributed shielding parameters for an arbitrary catchment geometry. Nonetheless, accounting only for surface skyline shielding (e.g., Codilean, 2006) without including its concurrent

influence on the effective attenuation length should be avoided.





## 6 Conclusions

The simplified model presented here for catchment-scale topographic shielding of incoming cosmic radiation highlights the two competing effects of slope and skyline shielding. As catchment relief increases, surface production rates are reduced due to increased skyline shielding. However, for shielded samples radiation is increasingly collimated, and for sloped surfaces oblique radiation pathways increase nuclide production at depth. Both of these effects lead to deeper effective mass attenuation lengths, which offset the reduction in surface production when inferring erosion rates from cosmogenic nuclide concentrations. At the catchment scale, the mean total effective shielding factor is one for a large range of catchment geometries, suggesting that topographic shielding corrections for catchment samples are generally not needed, and that applying commonly used topographic shielding algorithms leads to underestimation of true erosion rates by up to 20%. Although these corrections are typically small compared to other methodological uncertainties, they vary systematically with slope and relief. Consequently, misapplication of shielding correction factors could influence interpretations of relationships between topography and erosion rate.

## Competing interests

The author declares no conflict of interest.

## Acknowledgements

This project was supported by funding from National Science Foundation grant EAR-160814, and benefited from discussions with K. Whipple, A. Neely, and P. Bierman.

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





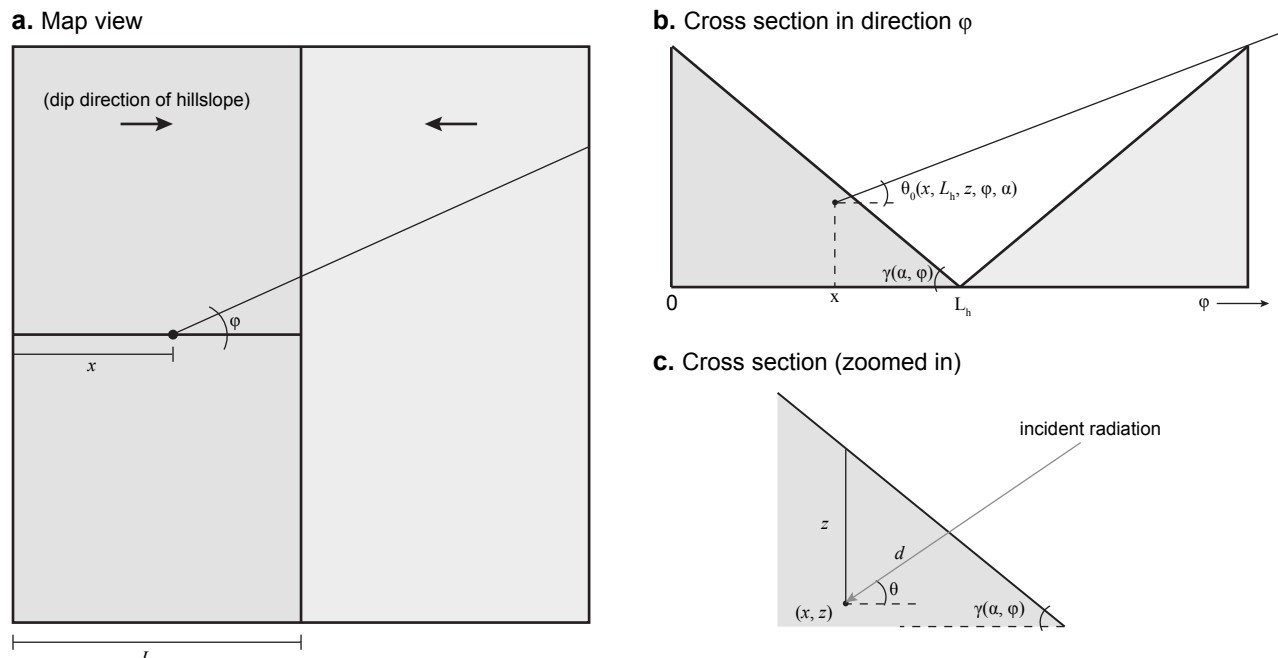

**Figure 1: Model catchment setup, showing (a) map view, (b) cross section along azimuthal angle $\varphi$ (note that $\gamma = \alpha$ for $\varphi = 0$), and (c) close up of hillslope cross section.**

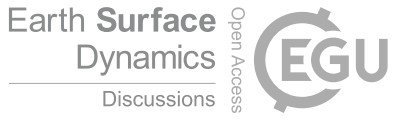



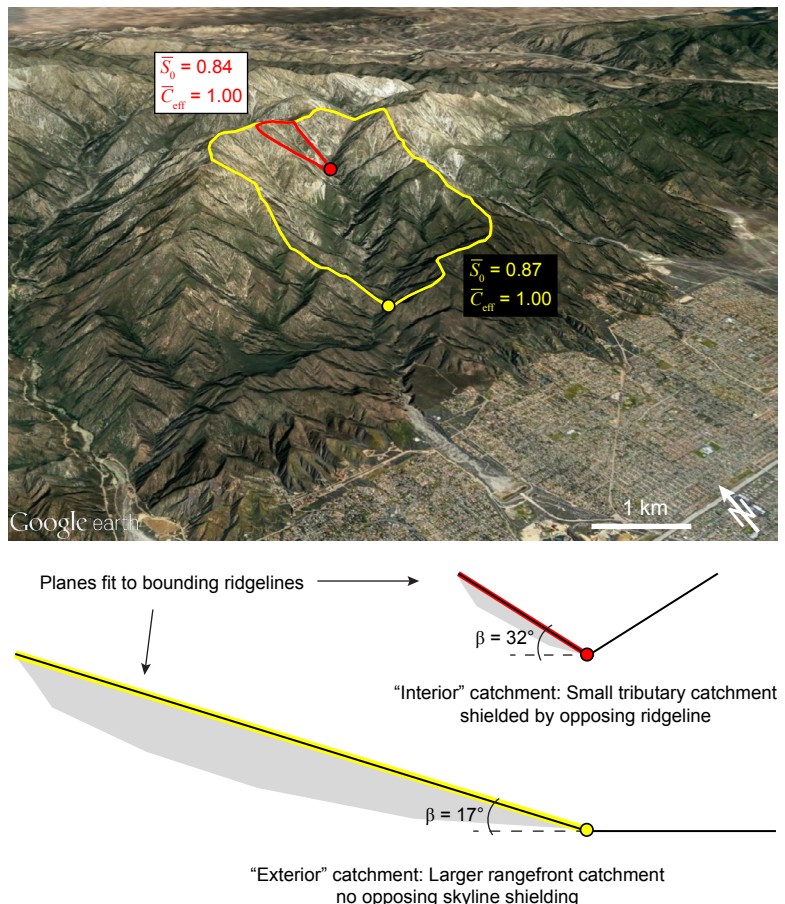

**Figure 2: Dipping catchment shielding geometry, illustrated using example from the San Gabriel Mountains, California, USA. Image is centered on 34.20°N, 117.61°W. Colored lines indicate planes fit through bounding ridgelines dipping at angle $\beta$. $\overline{S}_0$ indicates mean surface skyline shielding parameter calculated using algorithm of Codilean (2006), and $\overline{C}_{eff}$ indicates the mean total effective shielding factor calculated from the simplified catchment geometry.**



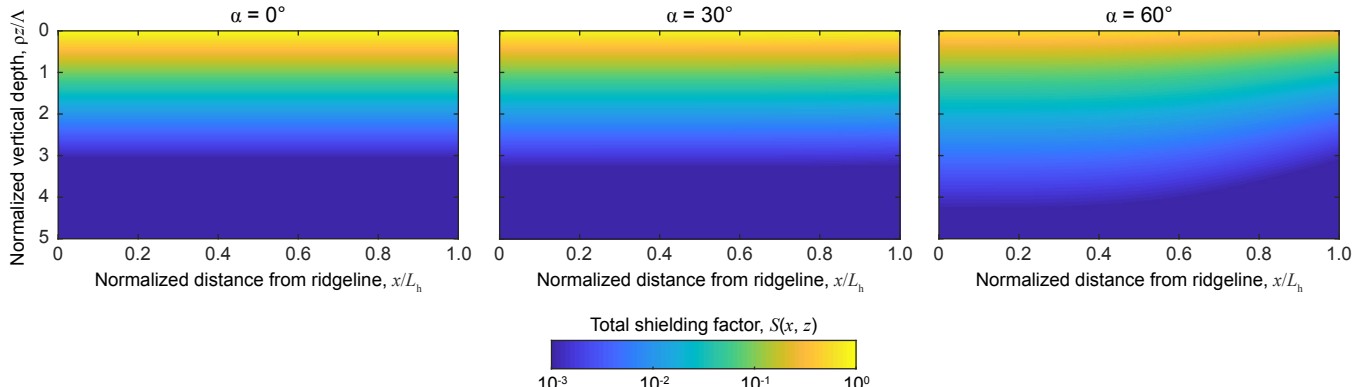

**Figure 3: Total shielding factor, $S(x, z)$, as a function of normalized vertical depth and distance from ridgeline for varying hillslope angle, $\alpha$.**





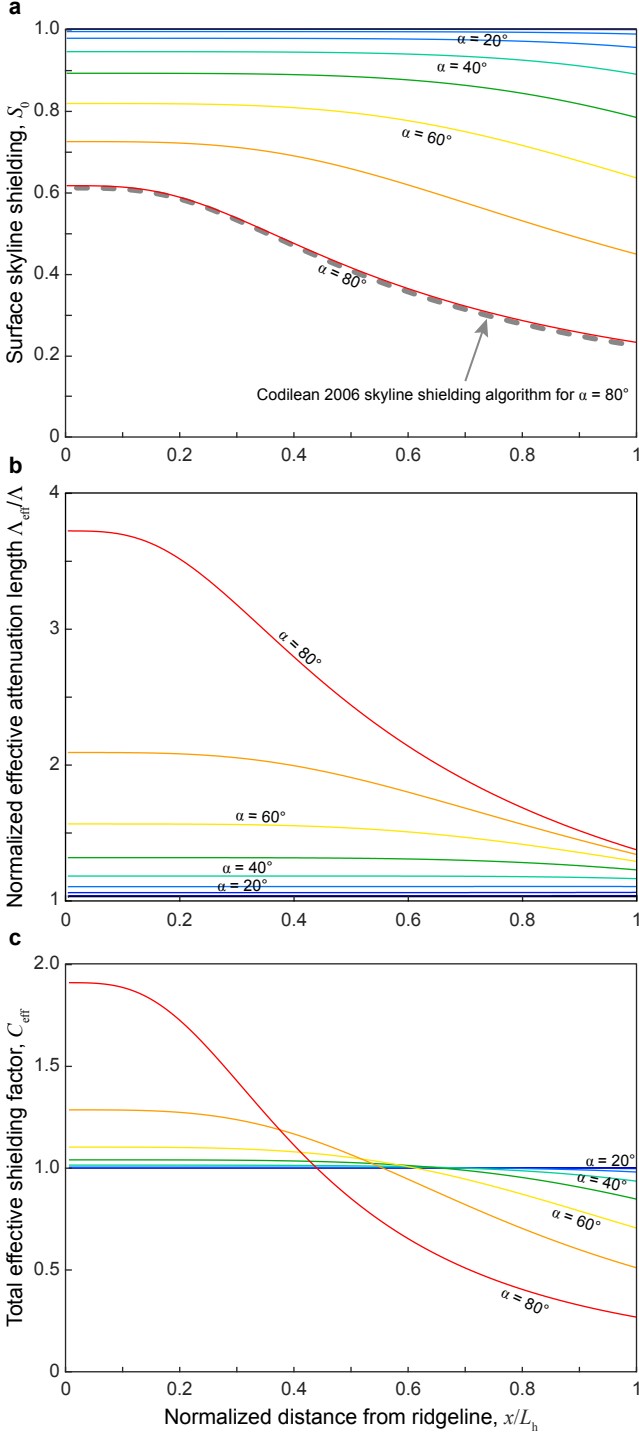

**Figure 4: Plots of (a) surface skyline shielding factor (b) normalized effective vertical attenuation length, and (c) total effective shielding factor as a function of distance from ridgeline for model runs with α = 0-80°. Dashed line in (a) indicates topographic shielding calculation using algorithm of Codilean (2006) applied to a digital elevation model of the case α = 80°.**



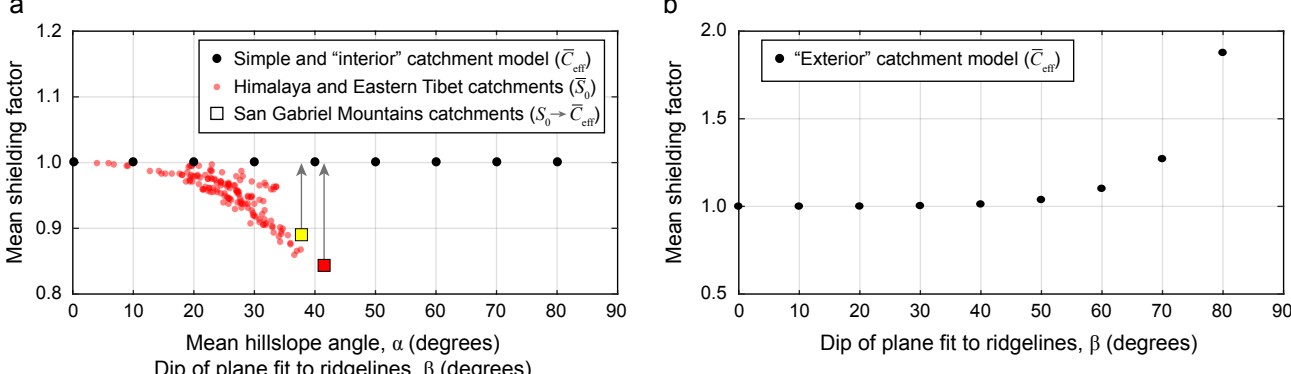

**Figure 5: Plots showing mean total shielding factor, $\overline{C}_{eff}$, for (a) simple horizontal catchment case (Fig. 1) for varying mean hillslope angle, $\alpha$, which is equivalent to the "interior" dipping catchment case as a function of catchment dip, $\beta$ (Fig. 2), and (b), the mean total shielding factor, $\overline{C}_{eff}$, for the "exterior" dipping catchment model as a function of catchment dip, $\beta$ (Fig. 2). Red points in (a) indicate relationship between the mean surface skyline shielding factor, $\overline{S}_0$, as a function of mean hillslope angle for compilation of catchment [10]Be data in the Himalaya and Eastern Tibet as reported by Scherler et al. (2017). Red and yellow squares indicate mean surface skyline factor, $\overline{S}_0$, calculated for example catchments from San Gabriel Mountains (Fig. 2). Arrows indicate difference between mean surface skyline shielding factor and mean total shielding factor, $\overline{C}_{eff}$.**