# Peer review of "Short Communication: Increasing vertical attenuation length of cosmogenic nuclide production on steep slopes negates topographic shielding corrections for catchment erosion rates"

_Earth Surface Dynamics, 2018_

## Referee Comment (RC1) · Anonymous Referee #1 · 15 Jul 2018

In this manuscript, DiBiase presents a thorough representation of cosmic ray shielding in eroding complex topography. By necessity, the manuscript covers much of the same ground as Dunne et al., 1999 and Gosse and Phillips, 2001 but extends those works by looking at the implications of geometry on the effective attenuation path length across an idealised landscape. This is a fascinating and provocative manuscript. The results suggest that the topographic shielding corrections that are often applied for cosmogenic nuclide derived erosion rates are not necessary. Taken at face value, this implies that previously reported denudation rates could be underestimated by up to 20%.

[Figure]

The manuscript is well written and, for the most part, clear. There are a few main points (all related) that I think require clarification: 1) the effect of foreshortening, 2) the role of changing surface production rates due to changing attenuation path length, and 3) the implications of defining attenuation path length as a vertical vector instead of its traditional definition as perpendicular to the surface.

1) Gosse and Phillips (2001) noted on pg 1521 that increasing effective attenuation length due to increasing surface slope is exactly offset by foreshortening. Is that not also the case here?

2) It is counterintuitive that the effective shielding factor can be greater than 1. In this model, this is due to the large increase in vertical attenuation length. Previous authors have noted that attenuation path length decreases on sloped surfaces due to increasingly oblique incidence angles reducing the intensity. This discrepancy should be addressed. On a similar topic, it is not clear is how production rates were dealt with here. For a give incoming flux, increasing the attenuation path length must decrease the near surface production rate as it implies fewer collisions per mass length. The implication is that as normalized effective attenuation length increases, the normalized effective surface production rate must decrease. This would offset the effect of increasing attenuation length (requiring a topographic shielding correction again). This could be treated as equivalent to foreshortening.

3) There is an important potential talking point here on how erosion/denudation is defined in cosmogenic nuclide studies. Both lowering rates (i.e. m My-1) and mass loss rates (i.e. t km-2 yr-1) tend to be based on 2D areas. This is in line with the definition of attenuation length presented here. However, it is not clear that this is the appropriate definition (of either erosion or attenuation) for the real world. A broader discussion around the implications of setting the attenuation path length to the vertical could be quite useful since previous authors rotate the coordinate system to determine attenuation path length perpendicular to the surface. The vertical definition makes sense since we tend to perform shielding calculations on a DEM and often define erosion

as a lowering rate. However, it seems unlikely that an 80° slope would be eroding vertically. In this case, using a vertical attenuation path length would result in an artificial increase in production rate (i.e. it would appear as less shielding, as found here). The 'true' surface area in this case is also probably the 3D surface area and erosion would be spread across a larger area (essentially the foreshortening argument applied to erosion). I recognise that this is a bit circular, but it highlights the need for a clearer explanation around coordinate definitions.

In summary, while there are some important ideas to clarify, this manuscript raises the very enticing idea that topographic shielding corrections are not needed for denudation studies. If this is indeed the case, then DiBiase will surely receive a whole-hearted 'thank you' from the cosmogenic nuclide community.

---

## Author Comment (AC1) · 20 Jul 2018

I thank Reviewer 1 for highlighting a few confusing geometrical complications to treating cosmogenic nuclide production on sloped surfaces, which emerge from differences in choice of coordinate system/frame of reference used by prior studies. I will work to clarify these points in the revised manuscript, but post responses to them now for discussion:

[Figure]

**Reviewer #1 Comment 1:** *"Gosse and Phillips (2001) noted on pg 1521 that increasing effective attenuation length due to increased surface slope is exactly offset by foreshortening. Is that not also the case here?"*

**Response:** As noted by Dunne et al. (1999), the competing factors described by Gosse and Phillips (2001) of changing effective attenuation length and foreshortening are already accounted for in the formulation of Equation 1 in the present manuscript (Equation 11 of Dunne et al., 1999). Specifically, these geometrical effects fall out by framing the problem using the mass attenuation length for collimated radiation, $\lambda$, and directly determining the mass length, d, along a ray path from the surface to a given position $(x, z)$ in the subsurface. The consequence of this explicit treatment of exponential attenuation in the ray path direction is nicely shown by Figure 5 and Equations 16 and 17 of Dunne et al., highlighting the difference between $\lambda$ and the more commonly used attenuation length, $\Lambda$.

**Reviewer #1 Comment 2:** *"It is counterintuitive that the effective shielding factor can be greater than 1. In this model, this is due to the large increase in vertical attenuation length. Previous authors have noted that attenuation path length decreases on sloped surfaces due to increasingly oblique incidence angles reducing the intensity. This discrepancy should be addressed. On a similar topic, it is not clear is how production rates were dealt with here. For a give incoming flux, increasing the attenuation path length must decrease the near surface production rate as it implies fewer collisions per mass length. The implication is that as normalized effective attenuation length increases, the normalized effective surface production rate must decrease. This would offset the effect of increasing attenuation length (requiring a topographic shielding correction again). This could be treated as equivalent to foreshortening."*

**Response:** Similar to the above response to point 1, the effects of changing attenuation path length and foreshortening are already accounted for in the model. The

counterintuitive result of shielding factors greater than 1 emerges entirely because of treating the problem in the vertical direction rather than normal to the surface – the cosine correction from slope-normal to vertical frame of reference quickly overcomes the shielding effect for sloped surfaces.

*Reviewer #1 Comment 3: "There is an important potential talking point here on how erosion/denudation is defined in cosmogenic nuclide studies. Both lowering rates (i.e. m My-1) and mass loss rates (i.e. t km-2 yr-1) tend to be based on 2D areas. This is in line with the definition of attenuation length presented here. However, it is not clear that this is the appropriate definition (of either erosion or attenuation) for the real world. A broader discussion around the implications of setting the attenuation path length to the vertical could be quite useful since previous authors rotate the coordinate system to determine attenuation path length perpendicular to the surface. The vertical definition makes sense since we tend to perform shielding calculations on a DEM and often define erosion as a lowering rate. However, it seems unlikely that an $80°$ slope would be eroding vertically. In this case, using a vertical attenuation path length would result in an artificial increase in production rate (i.e. it would appear as less shielding, as found here). The 'true' surface area in this case is also probably the 3D surface area and erosion would be spread across a larger area (essentially the foreshortening argument applied to erosion). I recognise that this is a bit circular, but it highlights the need for a clearer explanation around coordinate definitions."*

**Response:** Aside from the fact that the standard reporting of erosion rates is done in the vertical frame of reference, there is a grid-scale-dependence issue that arises when trying to quantify 3D area that is a significant limitation (e.g., Norton and Vanacker, 2009). This opens up a whole host of new challenges, and so treating the topographic shielding (and reporting of erosion rates) is best done in the vertical. Of course, this is not to say that surface process models might not be better cast in terms of horizontal retreat or surface-normal displacement. Rather, the conversion should be done in

comparison with model predictions, and not at the stage of reporting mass flux out of catchments or catchment-mean lowering rates. As I mention in the manuscript, the larger issue with very steep catchments is that the assumptions of steady lowering are more likely to be violated due to stochastic mass wasting processes.

---

## Referee Comment (RC2) · G. Balco (Referee) · 15 Aug 2018

Invited review of 'Increasing vertical attenuation length...'

1. Summary.

This is a thoroughly useful paper that does a great job of highlighting that if you have a complicated calculation, but you only do the easy complications and not the hard complications, you are probably getting the wrong answer. It's an important contribution to the field of erosion-rate measurement using cosmogenic nuclides that should be

published in approximately its present form. In the following text I have (i) some context for why this paper is valuable, and (ii) a few suggestions that I think would improve the paper.

In addition, I strongly encourage the author to check the calculations again. I don't see any reason to think anything is not correct, but I haven't attempted to fully replicate all the calculations. The main thing I haven't verified is the effect of the slope angle on the apparent attenuation length in the vertical direction – obviously, when you set the geometry up this way, $\Lambda$ in the vertical direction has to converge on infinity for a vertical cliff, so the principle is clearly correct, but it would be embarrassing to get this wrong and I suggest checking it carefully.

Finally, it does not appear at present that the computer code used to do the calculations is included with the paper or otherwise available (or, at least, I couldn't find it via the journal web interface). This is a deficiency and the author should correct it.

2. Details.

So, anyway, first, some context and an explanation of why this paper is valuable. Initially when cosmogenic-nuclide-based estimates of erosion rates began to be used, people mostly ignored the entire issue of topographic shielding, because it was evident that (i) it was basically negligible for most normal watersheds, and (ii) even if it wasn't, it was much less important than other assumptions, particularly the assumption of steady uniform erosion for a time much longer than $\lambda + \epsilon/\Lambda$. Subsequently, several people, including myself, realized that it was kind of a fun exercise in ARC/INFO, or whatever other GIS package, to compute surface shielding factors for entire watersheds, but never really did anything about it. Eventually Codilean (2006) wrote a paper about how to do this. Most discussion of this in published literature, e.g., Balco et al. (2008) and Mudd et al. (2016), included a sentence to the effect that, oh yeah, although you can calculate the topographic shielding effect on the surface production rate, there is also an effect on the subsurface attenuation that we haven't taken into account, the

unstated subtext being that yes, we really did read Dunne (1999), and calculating the surface shielding factor from a DEM is kind of fun, but then having to do a numerical integration for all depths in all pixels is a lot of work and not really that fun any more. So the present paper is completely correct to point out that the existing state of the art only does half the calculation.

Personally, I did eventually attempt to determine whether doing only half of the calculation was important using the following fairly simplistic analysis:

1. Assume steady erosion, spallation only, and a stable nuclide, such that for a steadily eroding surface $N = P\Lambda/\epsilon$, where $N$ is the surface nuclide concentration (atoms/g), $P$ is the surface production rate (atoms/g/yr), $\Lambda$ is an effective subsurface attenuation length (g/cm3/yr), and $\epsilon$ is the erosion rate (g/cm2/yr).

2. Topographic shielding reduces $P$ at the surface, but it also increases $\Lambda$ because the cosmic-ray flux becomes more collimated; if these offset each other equally then the overall effect on $N$ would be zero. To determine to what extent these offset each other, consider the nuclide concentration at a single point located in a cylindrical hole of varying depth, such that it has an apparent horizon that cuts off the cosmic-ray flux below some angle. This is basically what is in Figure 2 in Dunne (1999).

I just re-did this calculation using the numerical integration code in Balco (2014, Quat. Geochron.) and, of course, got the same results as Dunne. For this very simplified case (which, of course, ignores the fact that pixels in real watersheds aren't flat), the increase in $\Lambda$ only partially offsets the decrease in $P$ (compare Figs. 1 and 2 of Dunne), so increasing shielding does, in fact, decrease $N$ for a particular erosion rate. Thus, if one were to consider only the change in $P$ in computing erosion rates, one would get the wrong answer, but it wouldn't be that wrong. So, based on this analysis, I have not ever been very concerned in reading papers about erosion-rate estimates whether authors did or did not take topographic shielding into effect; I concluded that not including it, or alternatively including the effect on $P$ but not on $\Lambda$ following Codilean and Mudd,

would both be slightly wrong, but not very wrong; the correct answer would be some-where in the middle. I suspect this overall issue has so far not attracted very much attention because others have gone through some similar analysis and also concluded that it is not that big a deal, and there would be no significant overall science benefit in highlighting the fact that applying only the Codilean scheme is, technically, incorrect and misleading.

So the value of the present paper is that it shows that I, and perhaps others, are completely wrong in relying on a slightly-less-simple-than-Codilean-2006-but-still-too-simple analysis to conclude that this overall issue is probably not that important. This paper correctly points out that (i) increased attention to high-relief watersheds made possible by improved Be-10 measurements at low concentrations, as well as (ii) larger data sets and compiled databases that can be applied to quantifying relationships be-tween topographic metrics and erosion rates, mean that this issue is potentially impor-tant. Then this paper actually carries out the more complicated analysis, that no one else seems to have done, of the effect of variations in $\Lambda$ on nuclide concentrations for real(er) watershed geometries, and concludes that, in fact, considering only the effect on $P$ and not the effect on $\Lambda$ is significantly more inaccurate than considering neither. This is an important contribution that makes clear that everyone should have paid more attention to this issue.

Some comments that would improve the paper:

Page 1, line 15, in abstract. Using 'catchment mean effective shielding factor of one' is premature here and makes no sense to the reader, because you haven't defined it yet. Of course, you're not talking about the shielding factor as usually defined ($P/P_0$), you're talking about the effect of both production and attenuation ($P\Lambda/P_0\Lambda_0$). Instead you should say something like 'for flat catchments, the effect of increasing attenuation length due to shielding offsets the effect of decreasing surface production rate, result-ing in no change in surface nuclide concentrations in relation to the unshielded case.' Something like that, anyway. Clunkier but necessary to be more clear for purposes of

the abstract.

Page 2, line 6. "In general shown to be robust." Actually, I am only aware of the one paper by Granger that actually validates the method against sediment fluxes. Are there others? In any case, I'm not sure this is true at all. Maybe omit this.

Page 3, line 25. Why $\Lambda$ increases could be explained more clearly here. You never really say why this happens, which is twofold: for topographic shielding, you have excluded cosmic rays with oblique incidence angles that stop at relatively shallow depths (shallow in a vertical direction of course), leaving a higher fraction of near-vertical ray paths that stop at deeper depths, and then for sloping surfaces the vertical coordinate system means that points that are deep in the vertical coordinate aren't deep in a slope-normal direction any more.

Pages 3-4, end of section 2 and beginning of 3. I think it would be helpful here to introduce the simplest possible case, that is, stable nuclide, spallation only, as described above, where $N = P\Lambda/\epsilon$. That makes it clear that increasing $\Lambda$ has an offsetting effect on reducing $P$. As it is you just jump right into the complicated watershed-with-many-pixels case without really making the basic relationship clear.

Page 5, line 15-20. This is a little misleading as written, because of course for a flat surface the depth-dependence is not exactly exponential either, it's just a good approximation. How good depends on how you define the angular dependence of the flux. See the Argento paper.

Page 7, top, and in general throughout sections 2 and 3 as well. It is rather important to understanding all this that the reader really realizes that the z coordinate is always vertical, rather than surface-normal. There is nothing actually wrong with the paper here, but I suggest reminding the reader of this more times than seems necessary at first. For example, this would be a good place to remind the reader of this.

Page 8, line 20-ish. For muons, the penetration depth is so long and the associated

time-to-equilibrium is likewise so long that there is really no plausible scenario in which the steady state assumption is ever correct. Thus, calculating the surface nuclide concentration due to muon production for a given erosion rate is pretty wildly speculative to begin with. Like, for rapidly eroding catchments where a significant fraction of production is by muons, you can get almost factor-of-two differences between, for example, assuming steady-state erosion and assuming that the erosion rate sped up recently. Possibly worth pointing this out here.

Page 8, line 30. This is funny because it is written from the perspective of people who care about steep, rapidly eroding basins, where one worries that the steady-state assumption is wrong because of stochastic landslides/slope failures. In contrast, if you are a person who cares about slowly eroding basins, instead you worry that the steady-state assumption is wrong because there is no way that erosion rates haven't been unsteady due to glacial-interglacial-scale climate changes. You should probably mention both things here.

Page 8, line 30. One shouldn't say "should" in papers (apparently only in reviews). Just state the facts — "The analysis here shows that accounting only for surface skyline shielding yields incorrect results." Let the reader decide what to do about it. The conclusions on the next page are much better in this regard.

3. One comment about the other review.

The issue of "foreshortening" commonly comes up in this discussion as a point of ambiguity because most people with an Earth science background are used to thinking about this effect in the context of remote sensing or radiative heating, in which a finite amount of incident radiation is spread out over a larger area when the incidence angle is lower. The key difference is that cosmogenic-nuclide production happens within a volume, not on a surface, so the radiation-incident-on-a-surface model is not the right way to think about this, and the treatment of foreshortening that would be used in that context is not applicable here. The clearest discussion of this issue is in Dunne, and

the present paper follows Dunne and gets it right.

---

## Author Comment (AC2) · 16 Aug 2018

I thank Reviewer 2 (G. Balco) for the constructive review, and will incorporate his feedback into the revised manuscript. For now, I cleaned up the Matlab code used to run the model and generate Figs. 3-5 and will include this as a supplement to the revised manuscript (GitHub repository can be found here: https://github.com/romandibiase/catchment-shielding).

[Figure]

2018.

---

## Referee Comment (RC3) · G. Balco (Referee) · 21 Aug 2018

This is just to clarify the comment in the Scherler review, at the top of page C2:

"That basically means that we assume that all particles approaching the Earth's surface follow trajectories that are normal to the geoid."

As this is written, it sounds incorrect – in the DiBiase paper, as in other work, the cosmic-ray flux is assumed to be distributed around the upper hemisphere such that the intensity is proportional to the cosine of the zenith angle taken to a power. So it is

true that the intensity (particles per time, per area normal to the particle direction, per unit solid angle) of cosmic-ray particles from the vertical direction (e.g., "normal to the geoid") is higher than the intensity of particles arriving from lower angles. But incident particles arrive from all directions in the upper hemisphere.

More generally, let me try to clarify the frame-of-reference issue a bit more. The important thing is that there are two parts to the calculation: computing surface and subsurface production rates, and then using that information to relate erosion rates to surface nuclide concentrations. Obviously, the production rate at a particular surface or subsurface location is the same no matter whether the reference frame is vertical or slope-normal. The reference frame only becomes important when it is necessary to compute the relation between erosion rate and surface nuclide concentration, because you need to define the direction in which erosion is taking place and therefore in which you are integrating production rates. A vertical reference frame, of course, is nearly always used in erosion rate applications because in this reference frame, each pixel has the same area, so it is not necessary to consider different pixel areas in averaging the nuclide concentration from a group of pixels with different slopes. A slope-normal reference frame, as far as I am aware, has only been used in a few papers (like the Ward and Anderson example) that are mostly concerned with cliff retreat rates, because if the cliff is vertical and the reference frame is vertical, then cliff retreat is effectively mixing subsurface and surface sediment, which would violate basic assumptions of watershed erosion-rate calculations. Also, as shown by Masarik, there are other effects besides shielding that are important in computing production rates under near-vertical surfaces, so the exponential approximation doesn't really work very well anyway.

The present paper chooses the vertical reference frame, which is likely most appropriate anyway because, as Scherler notes, most watersheds do not have vertical cliffs, so the fact that cliff retreat violates the usual assumptions of erosion-rate calculations is not relevant. It is true that the attenuation length computed in a vertical direction underneath a dipping surface is totally different from the attenuation length computed

normal to the surface. This is confusing, because the same symbols are used in various papers for both things, so it is necessary to be clear how the attenuation length is defined, and I agree that this could be pointed out more often in the paper.

In the context of computing an erosion rate, however, the difference between the slope-normal attenuation length and the vertical attenuation length is just the same as the difference between the slope-normal erosion rate and the vertical erosion rate, so if we consider the basic relation $N = P\Lambda/\epsilon$, then (given several assumptions) changing to a slope-normal erosion rate and slope-normal $\Lambda$ doesn't affect $P$, but makes both $\Lambda$ and $\epsilon$ smaller by a factor of the cosine of the angle, so the nuclide concentration is unchanged.

This also highlights why cliffs are trouble, because as the slope approaches vertical, the slope-normal erosion rate approaches zero, which is probably not what you want.

---

## Short Comment (SC1) · 21 Aug 2018

Invited Review of esurf-2018-48

The contribution by R. DiBiase addresses a methodological issue in calculating catchment-averaged erosion rates from cosmogenic nuclide concentrations in river sediments that has been largely overlooked in the past. The paper is well written, the figures are informative and relevant references are given. Overall, I agree with the previous reviewers that this is a great paper that should be published after some

(minor) clarifying revisions.

My main point is in line with what reviewer #1 already mentioned. I think it is important to emphasize that the reference frame, in which the attenuation length increases with increasing slope angle, is vertical with respect to the geoid and not the surface itself. That basically means that we assume that all particles approaching the Earth's surface follow trajectories that are normal to the geoid. While this assumption appears reasonable for hillslope angles <30° or so, to me it appears unreasonable for very steep hillslope angles, where the described effect is most pronounced. When standing in front of a rock face that is inclined 60° or more, I guess that most people would think the rock wall retreats and not that it lowers. The resulting particle trajectories would thus be less steeply inclined with respect to the surface and the effective attenuation length would not be that large. As a result, the shielding effect would likely be significant, hence lowering the surface production rates; but there would be no counter-acting effect due to increasing attenuation length. Dylan Ward and Bob Anderson, for example, looked at steep hillslopes in glaciated landscapes and assume slope-normal trajectories (Ward and Anderson, 2010, Earth Surf. Process. Landforms 36, 495-512). I think this is an important point that needs to be better exposed in the beginning and discussed later on.

My second point is related: is it meaningful to show on Figure 4, curves for inclinations up to 80°? I would argue that there hardly exist catchments with mean hillslope angles of >40°. Such angles may exist locally, but are they relevant for the problem that you discuss? One solution could be to have the y-axis in log scaling, to emphasize the curves with angles <40°, which currently are hard to decipher. As you rightfully note in your discussion, the effect of topographic shielding is small in most cases. All the curves >40° are thus steering the readers attention towards cases that actually don't matter.

A few more minor points:

P2, Line 18: You cite Norton and Vanacker (2009), but you don't discuss the main point of their paper in any detail later one. I think you should, because they propose that topographic shielding measured from coarse DEMs may underestimate the actual shielding. If that were true, does it mean that, after taking different attenuation lengths into account, there might still be a net shielding effect?

P3, Line 28: Probably here you could mention more explicitly the assumed particle trajectory. You actually say "vertical depth below the surface", but that's ambiguous. Vertical with respect to the surface or the geoid?

P4, Line 12: Mention here already if the model valley is inclined?

P5, Line 11: How good is this approximation?

P8, Line 17: The factor 3 emerges only for hillslopes >80°. I think it would be better here to refer to commonly observed hillslope angles, given the title of this chapter, and not extreme cases.

Figure 5: I'm curious whether it is ok to refer to mean hillslope angles? Pixel-based hillslope angles are often measured using the steepest descent algorithm. In other words, this algorithm will give you always the maximum slope angle possible. Is that the one you want to have for inferring attenuation length effects? Or would you rather want to refer to hillslope angles measured by fitting a plane to each pixel and its surrounding neighbors, or something like this?

Dirk Scherler

---

## Author Response (AR1)

9/6/2018

Dear Dr. Willenbring,

Thank you for considering publication in *Earth Surface Dynamics* the manuscript "Short Communication: Increasing vertical attenuation length of cosmogenic nuclide production on steep slopes negates topographic shielding corrections for catchment erosion rates". I appreciate the constructive feedback from three reviews, and have revised the manuscript to address reviewer comments, as outlined in detail below (reviewer comments are *italicized*). I hope you will find the revised manuscript ready for publication.

Sincerely,
Roman DiBiase

**Reviewer #1 comments (Anonymous)**

1) *Gosse and Phillips (2001) noted on pg 1521 that increasing effective attenuation length due to increased surface slope is exactly offset by foreshortening. Is that not also the case here?*

   **Response:** As noted by Dunne et al. (1999), the competing factors described by Gosse and Phillips (2001) of changing effective attenuation length and foreshortening are already accounted for in the formulation of Equation 1 in the present manuscript (Equation 11 of Dunne et al., 1999). Specifically, these geometrical effects fall out by framing the problem using the mass attenuation length for collimated radiation, $\lambda$, and directly determining the mass length, $d$, along a ray path from the surface to a given position $(x, z)$ in the subsurface. The consequence of this explicit treatment of exponential attenuation in the ray path direction is nicely shown by Figure 5 and Equations 16 and 17 of Dunne et al. (1999), highlighting the difference between $\lambda$ and the more commonly used attenuation length, $\Lambda$.

   I also include the helpful explanation of the foreshortening issue by G. Balco in his review below:

   > "The issue of "foreshortening" commonly comes up in this discussion as a point of ambiguity because most people with an Earth science background are used to thinking about this effect in the context of remote sensing or radiative heating, in which a finite amount of incident radiation is spread out over a larger area when the incidence angle is lower. The key difference is that cosmogenic-nuclide production happens within a volume, not on a surface, so the radiation-incident-on-a-surface model is not the right way to think about this, and the treatment of foreshortening that would be used in that context is not applicable here. The clearest discussion of this issue is in Dunne, and the present paper follows Dunne and gets it right."

**2)** *It is counterintuitive that the effective shielding factor can be greater than 1. In this model, this is due to the large increase in vertical attenuation length. Previous authors have noted that attenuation path length decreases on sloped surfaces due to increasingly oblique incidence angles reducing the intensity. This discrepancy should be addressed. On a similar topic, it is not clear is how production rates were dealt with here. For a give incoming flux, increasing the attenuation path length must decrease the near surface production rate as it implies fewer collisions per mass length. The implication is that as normalized effective attenuation length increases, the normalized effective surface production rate must decrease. This would offset the effect of increasing attenuation length (requiring a topographic shielding correction again). This could be treated as equivalent to foreshortening.*

**Response:** Similar to the above response to point **1**, the effects of changing attenuation path length and "foreshortening" are already accounted for in the model. The counterintuitive result of shielding factors greater than 1 emerges entirely because of treating the problem in the vertical direction rather than normal to the surface – the cosine correction from slope-normal to vertical frame of reference quickly overcomes the shielding effect for sloped surfaces. To clarify this transition, I now include a new figure highlighting the change in normalized production rate as a function of depth for a 60 degree sloping surface (Fig. 5):

[Figure]

"Figure 5: Plot of normalized production rate relative to horizontal unshielded surface as a function of normalized vertical depth for a 60° slope with no additional skyline shielding. Near the surface, production rates are decreased due to slope shielding of incoming cosmic radiation; however, production rates at depth increase relative to the unshielded case due to additional radiation along shorter oblique pathways (Fig. 1c)."

**3)** *There is an important potential talking point here on how erosion/denudation is defined in cosmogenic nuclide studies. Both lowering rates (i.e. m My-1) and mass loss rates (i.e. t km-2 yr-1) tend to be based on 2D areas. This is in line with the definition of attenuation length presented here. However, it is not clear that this is the appropriate definition (of either erosion or attenuation) for the real world. A broader discussion around the implications of setting the attenuation path length to the vertical could be*

*quite useful since previous authors rotate the coordinate system to determine attenuation path length perpendicular to the surface. The vertical definition makes sense since we tend to perform shielding calculations on a DEM and often define erosion as a lowering rate. However, it seems unlikely that an 80° slope would be eroding vertically. In this case, using a vertical attenuation path length would result in an artificial increase in production rate (i.e. it would appear as less shielding, as found here). The 'true' surface area in this case is also probably the 3D surface area and erosion would be spread across a larger area (essentially the foreshortening argument applied to erosion). I recognise that this is a bit circular, but it highlights the need for a clearer explanation around coordinate definitions.*

**Response:** I now include a new paragraph at the beginning of section 3 that clarifies and justifies the use of a vertical reference frame:

> "Throughout the analysis below, both the effective mass attenuation length, $\Lambda_{eff}$, and erosion rate, $E$, are defined in the vertical, rather than slope-normal direction. The vertical (with respect to the geoid) reference frame was chosen for three reasons. First, most studies report erosion rate as a vertical lowering rate and assume primarily vertical exhumation pathways. Second, treatment of slope-normal processes introduces a grid-scale dependence of erosion and shielding calculations that varies with topographic roughness (Norton and Vanacker, 2009). Third, for the case of uniform erosion rate, the resulting shielding calculations do not depend on the choice of reference frame, as long as the orientation of $\Lambda_{eff}$ and $E$ are defined similarly." (Page 4, Line 26-31)

Additionally, I added a sentence to the discussion highlighting the limitations of this approach for treating landscapes dominated by cliff retreat:

> "For steep catchments with spatially variable quartz content or erosion rate, direct calculation of shielding at depth is likely needed to calculate the spatially distributed total effective shielding parameter. In particular, shielding calculations in landscapes dominated by cliff retreat are poorly suited for treatment in a vertical reference frame (e.g., Ward and Anderson, 2011)." (Page 8, Line 26-29)

**Reviewer #2 comments (G. Balco)**

4) *I strongly encourage the author to check the calculations again. I don't see any reason to think anything is not correct, but I haven't attempted to fully replicate all the calculations. The main thing I haven't verified is the effect of the slope angle on the apparent attenuation length in the vertical direction – obviously, when you set the geometry up this way, in the vertical direction has to converge on infinity for a vertical cliff, so the principle is clearly correct, but it would be embarrassing to get this wrong and I suggest checking it carefully.*

**Response:** I went through and checked all calculations/equations again, and cleaned up a few typos (the code and results are unchanged). The inclusion of the Matlab code as a supplement should help others reproduce these calculations.

**5)** *It does not appear at present that the computer code used to do the calculations is included with the paper or otherwise available (or, at least, I couldn't find it via the journal web interface). This is a deficiency and the author should correct it.*

**Response:** I agree – this was an oversight on my part, and I have posted the Matlab code on GitHub and include a script to generate Figs. 3,4, and 6 as a supplement to this manuscript (referred to in the acknowledgements).

**6)** *Page 1, line 15, in abstract. Using 'catchment mean effective shielding factor of one' is premature here and makes no sense to the reader, because you haven't defined it yet. Of course, you're not talking about the shielding factor as usually defined ($P/P_0$), you're talking about the effect of both production and attenuation ($P\Lambda/P_0\Lambda_0$). Instead you should say something like 'for flat catchments, the effect of increasing attenuation length due to shielding offsets the effect of decreasing surface production rate, resulting in no change in surface nuclide concentrations in relation to the unshielded case.' Something like that, anyway. Clunkier but necessary to be more clear for purposes of the abstract.*

**Response:** The abstract has been reworded for clarity and now includes a definition of the total effective shielding factor:

> "The most common method for calculating topographic shielding accounts only for the reduction of nuclide production rates due to shielding at the surface, leading to catchment-mean corrections of up to 20% in steep landscapes, and makes the simplifying assumption that the effective mass attenuation length for a given nuclide production mechanism is spatially uniform. Here I evaluate the validity of this assumption using a simplified catchment geometry with mean slopes ranging from 0° to 80° to calculate the spatial variation in surface skyline shielding, effective mass attenuation length, and the total effective shielding factor, defined as the ratio of the shielded surface nuclide concentration to that of an unshielded horizontal surface. For flat catchments (i.e., uniform elevation of bounding ridgelines), the effect of increasing vertical attenuation length as a function of hillslope angle and skyline shielding exactly offsets the effect of decreasing surface production rate, indicating that no topographic shielding correction is needed when calculating catchment-mean vertical erosion rates. For dipping catchments (as characterized by a plane fit to the bounding ridgelines), the catchment-mean surface nuclide concentrations are also equal to that of an unshielded horizontal surface, except for cases of extremely steep range-front catchments, where the surface nuclide concentrations are counterintuitively higher than the unshielded case due to added production from oblique cosmic ray paths at depth." (Page 1 Line 9-20)

**7)** *Page 2, line 6. "In general shown to be robust." Actually, I am only aware of the one paper by Granger that actually validates the method against sediment fluxes. Are there others? In any case, I'm not sure this is true at all. Maybe omit this.*

**Response:** I've omitted this admittedly vague statement.

**8)** *Page 3, line 25. Why $\Lambda$ increases could be explained more clearly here. You never really say why this happens, which is twofold: for topographic shielding, you have excluded cosmic rays with oblique incidence angles that stop at relatively shallow depths (shallow*

*in a vertical direction of course), leaving a higher fraction of near-vertical ray paths that stop at deeper depths, and then for sloping surfaces the vertical coordinate system means that points that are deep in the vertical coordinate aren't deep in a slope-normal direction any more.*

**Response:** In section 2, my main focus is introducing the general theoretical framework for the treatment of shielding at depth. The effect of surface slope does not enter until the model geometry setup in section 3, so I think it is perhaps premature to discuss the influence of the vertical coordinate system on $\Lambda$ up front. Instead, I explain why $\Lambda$ increases in the presentation of model results (Page 7, Line 14-26).

9) *Pages 3-4, end of section 2 and beginning of 3. I think it would be helpful here to introduce the simplest possible case, that is, stable nuclide, spallation only, as described above, where N = P$\Lambda$/ε. That makes it clear that increasing $\Lambda$ has an offsetting effect on reducing P. As it is you just jump right into the complicated watershed-with-manypixels case without really making the basic relationship clear.*

**Response:** This is a good idea. I now briefly summarize the simple case to highlight how $P$ and $\Lambda$ contribute to the surface nuclide concentration:

> "The importance of accounting for both changes in surface production rate, $P$, and changes in the effective mass attenuation length, $\Lambda_{eff}$, is illustrated by the analytical solution for nuclide concentration, $C$, measured on a steadily-eroding surface for a stable nuclide with an exponential decrease of production rate with depth:
> $$C = P\Lambda_{eff}/E \tag{5}$$
> where $E$ is erosion rate (g cm$^{-2}$ yr$^{-1}$) (Lal, 1991). From Eq. (5) it is clear that increasing $\Lambda_{eff}$ counters the effect of decreasing $P$ in determining the surface nuclide concentration (or alternatively for inferring erosion rate)." (Page 4, Line 9-14)

10) *Page 5, line 15-20. This is a little misleading as written, because of course for a flat surface the depth-dependence is not exactly exponential either, it's just a good approximation. How good depends on how you define the angular dependence of the flux. See the Argento paper.*

**Response:** Good point. I revised this section to make it clear that 1) the exponential decrease for unshielded horizontal surfaces is expected by Equation 1; and 2) any deviation from exponential production with depth will lead to inaccuracies using the analytical solution for steady erosion:

> "Although spallogenic production of cosmogenic nuclides following Eq. (1) is well-described by an exponential decrease with depth for horizontal unshielded surfaces, this is not true in general for shielded samples (Dunne et al., 1999). The effective vertical mass attenuation length, $\Lambda_{eff}(x)$, is approximated by the vertical depth below the surface at which the shielding factor is 5% of the surface shielding (i.e., 3 e-folding lengths) such that:
> $$S(x, \frac{3\Lambda_{eff}(x)}{\rho}) = 0.05S(x, 0). \tag{9}$$

If nuclide production as a function of depth deviates from an exponential decline, it is inaccurate to use the analytical relationship between surface sample concentration, $C(x)$ (atoms g$^{-1}$), and steady-state vertical erosion rate, $E$ (g cm$^{-2}$ yr$^{-1}$), typically applied to eroding samples" (Page 5, Line 26 to Page 6, Line 4)

11) *Page 7, top, and in general throughout sections 2 and 3 as well. It is rather important to understanding all this that the reader really realizes that the z coordinate is always vertical, rather than surface-normal. There is nothing actually wrong with the paper here, but I suggest reminding the reader of this more times than seems necessary at first. For example, this would be a good place to remind the reader of this.*

**Response:** Good suggestion. I have added clarification of the vertical frame of reference for depth, *z*, throughout the manuscript. I also added a paragraph at the beginning of Section 3 to emphasize this:

"Throughout the analysis below, both the effective mass attenuation length, $\Lambda_{eff}$, and erosion rate, $E$, are defined in the vertical, rather than slope-normal direction. The vertical (with respect to the geoid) reference frame was chosen for three reasons. First, most studies report erosion rate as a vertical lowering rate and assume primarily vertical exhumation pathways. Second, treatment of slope-normal processes introduces a grid-scale dependence of erosion and shielding calculations that varies with topographic roughness (Norton and Vanacker, 2009). Third, for the case of uniform erosion rate, the resulting shielding calculations do not depend on the choice of reference frame, as long as the orientation of $\Lambda_{eff}$ and $E$ are defined similarly." (Page 4, Line 26-31)

12) *Page 8, line 20-ish. For muons, the penetration depth is so long and the associated time-to-equilibrium is likewise so long that there is really no plausible scenario in which the steady state assumption is ever correct. Thus, calculating the surface nuclide concentration due to muon production for a given erosion rate is pretty wildly speculative to begin with. Like, for rapidly eroding catchments where a significant fraction of production is by muons, you can get almost factor-of-two differences between, for example, assuming steady-state erosion and assuming that the erosion rate sped up recently. Possibly worth pointing this out here.*

**Response:** Good suggestion. This complication is now addressed:

"…the assumption of steady lowering is likely to be increasingly inappropriate for rapidly eroding landscapes characterized by a significant contribution of muonogenic production or slowly-eroding landscapes where $^{10}$Be concentrations integrate over glacial-interglacial climate cycles." (Page 9, Line 16-18)

13) *Page 8, line 30. This is funny because it is written from the perspective of people who care about steep, rapidly eroding basins, where one worries that the steady-state assumption is wrong because of stochastic landslides/slope failures. In contrast, if you are a person who cares about slowly eroding basins, instead you worry that the steady-state assumption is wrong because there is no way that erosion rates haven't been unsteady due to glacial-interglacial-scale climate changes. You should probably mention both things here.*

**Response:** Good suggestion. I revised the paragraph in question to address this complication:

> "…the assumption of steady lowering is likely to be increasingly inappropriate for rapidly eroding landscapes characterized by a significant contribution of muonogenic production or slowly-eroding landscapes where [10]Be concentrations integrate over glacial-interglacial climate cycles." (Page 9, Line 16-18)

14) *Page 8, line 30. One shouldn't say "should" in papers (apparently only in reviews). Just state the facts — "The analysis here shows that accounting only for surface skyline shielding yields incorrect results." Let the reader decide what to do about it. The conclusions on the next page are much better in this regard.*

**Response:** Thanks—I should have caught this earlier. Fixed:

> "Nonetheless, in all cases accounting only for surface skyline shielding (e.g., Codilean, 2006) without including its concurrent influence on the effective attenuation length yields incorrect results." (Page 9, Line 20-22)

**Reviewer #3 comments (D. Scherler)**

15) *My main point is in line with what reviewer #1 already mentioned. I think it is important to emphasize that the reference frame, in which the attenuation length increases with increasing slope angle, is vertical with respect to the geoid and not the surface itself. That basically means that we assume that all particles approaching the Earth's surface follow trajectories that are normal to the geoid. While this assumption appears reasonable for hillslope angles <30° or so, to me it appears unreasonable for very steep hillslope angles, where the described effect is most pronounced. When standing in front of a rock face that is inclined 60° or more, I guess that most people would think the rock wall retreats and not that it lowers. The resulting particle trajectories would thus be less steeply inclined with respect to the surface and the effective attenuation length would not be that large. As a result, the shielding effect would likely be significant, hence lowering the surface production rates; but there would be no counter-acting effect due to increasing attenuation length. Dylan Ward and Bob Anderson, for example, looked at steep hillslopes in glaciated landscapes and assume slope-normal trajectories (Ward and Anderson, 2010, Earth Surf. Process. Landforms 36, 495-512). I think this is an important point that needs to be better exposed in the beginning and discussed later on.*

**Response:** I now include a new paragraph at the beginning of section 3 that clarifies and justifies the use of a vertical reference frame:

> "Throughout the analysis below, both the effective mass attenuation length, $\Lambda_{eff}$, and erosion rate, $E$, are defined in the vertical, rather than slope-normal direction. The vertical (with respect to the geoid) reference frame was chosen for three reasons. First, most studies report erosion rate as a vertical lowering rate and assume primarily vertical exhumation pathways. Second, treatment of slope-normal processes introduces a grid-scale dependence of erosion and shielding calculations that varies with topographic

roughness (Norton and Vanacker, 2009). Third, for the case of uniform erosion rate, the resulting shielding calculations do not depend on the choice of reference frame, as long as the orientation of $\Lambda_{eff}$ and $E$ are defined similarly." (Page 4, Line 26-31)

Additionally, I added a sentence to the discussion highlighting the limitations of this approach for treating landscapes dominated by cliff retreat:

> "For steep catchments with spatially variable quartz content or erosion rate, direct calculation of shielding at depth is likely needed to calculate the spatially distributed total effective shielding parameter. In particular, shielding calculations in landscapes dominated by cliff retreat are poorly suited for treatment in a vertical reference frame (e.g., Ward and Anderson, 2011)." (Page 8, Line 26-29)

**16)** *My second point is related: is it meaningful to show on Figure 4, curves for inclinations up to 80°? I would argue that there hardly exist catchments with mean hillslope angles of >40°. Such angles may exist locally, but are they relevant for the problem that you discuss? One solution could be to have the y-axis in log scaling, to emphasize the curves with angles <40°, which currently are hard to decipher. As you rightfully note in your discussion, the effect of topographic shielding is small in most cases. All the curves >40° are thus steering the readers attention towards cases that actually don't matter.*

**Response:** Although it is true that few catchments exist with slopes >50-60°, I think it is important to highlight the extreme cases to emphasize: 1) the catchment shielding correction is not simply smaller than previously assumed, but cancels out entirely for most watersheds; and 2) the spatial variability of factors that control surface nuclide concentration on steep hillslopes. I also find it helpful to better intuit the model behavior by including a wide range of slopes.

I tried changing the y-axis on the plots in Figure 4 to a logarithmic scale, but this does not actually help much the visualization as there is only a factor of 4-5 variation in the parameters being plotted.

**17)** *P2, Line 18: You cite Norton and Vanacker (2009), but you don't discuss the main point of their paper in any detail later one. I think you should, because they propose that topographic shielding measured from coarse DEMs may underestimate the actual shielding. If that were true, does it mean that, after taking different attenuation lengths into account, there might still be a net shielding effect?*

**Response:** I now include an additional citation to Norton and Vanacker in discussing the potential influence of rough topography:

> "However, while not entirely transferable to arbitrarily rough topography (e.g., Norton and Vanacker, 2009), Fig. 4c suggests that for slopes less than 40°, the total effective shielding factor does not vary significantly across the hillslope." (Page 8, Lines 25-26)

Note that the slopes measured on coarse DEMs are also typically lower than those of high-resolution DEMs, such that the increase in attenuation length will be

commensurately smaller. It is not straightforward to model the effects of surface roughness, but my intuition is that these effects will cancel out for rough surfaces and lead to similar interpretations of (vertical) erosion rate.

**18)** *P3, Line 28: Probably here you could mention more explicitly the assumed particle trajectory. You actually say "vertical depth below the surface", but that's ambiguous. Vertical with respect to the surface or the geoid?.*

**Response:** This is a good point to make explicitly. At this point in the manuscript, I have not yet introduced complications associated with sloped surfaces. I add a note about vertical exhumation pathways on Page 6:

> "$t_{surface} = t_0 + \rho z_0/E$ is the time it takes for a rock parcel to travel from depth $z_0$ to the surface (assuming a vertical exhumation pathway)." (Page 6, Line 13-14)

**19)** *P4, Line 12: Mention here already if the model valley is inclined?*

**Response:** I revised this sentence to emphasize the model geometry:

> "Because the ridgelines have uniform elevation, there is no net dip to the catchment; the effect of valley inclination will be assessed in Section 3.3." (Page 5, Line 3-4)

**20)** *P5, Line 11: How good is this approximation?*

**Response:** It depends on the application, and so it difficult to state concisely here. Mainly, I use this as a way to frame the need for characterizing the effective mass attenuation length numerically according to Eq. (10).

**21)** *P8, Line 17: The factor 3 emerges only for hillslopes >80°. I think it would be better here to refer to commonly observed hillslope angles, given the title of this chapter, and not extreme cases.*

**Response:** The factor of 3 and 30% values are both for extreme cases – I added a sentence to highlight a more typical range of effective attenuation length increase due to collimation and slope-effects:

> "However, the magnitude of changes in the effective mass attenuation length due to shielding-induced collimation is at most 30% (Dunne et al., 1999), compared to the potentially factor of 3 or more increase due to shorter oblique radiation pathways on very steep slopes (Fig. 1c; Fig. 4b). For hillslope gradients commonly observed in cosmogenic nuclide studies of steep landscapes (30-40°), the increase in effective mass attenuation length due to shielding-induced collimation and slope effects are 2-5% and 6-15%, respectively (Dunne et al., 1999; Fig. 4b). The dependence of $\Lambda$ on atmospheric depth, which is typically not accounted for in catchment erosion studies, is minor (<10% for extreme case of catchment with 4 km of relief (Marrero et al., 2016)) compared to the above slope effect for most landscapes." (Page 9, Line 4-11)

**22)** *Figure 5: I'm curious whether it is ok to refer to mean hillslope angles? Pixel-based hillslope angles are often measured using the steepest descent algorithm. In other words, this algorithm will give you always the maximum slope angle possible. Is that the one you want to have for inferring attenuation length effects? Or would you rather want to refer to hillslope angles measured by fitting a plane to each pixel and its surrounding neighbors, or something like this?*

**Response:** For catchment-mean hillslope angles, there is not too much difference between measuring local slope along a steepest descent path versus fitting a plane to a local neighborhood. The biggest difference in resulting values is related to the difference in the scale of measurement (i.e., calculating over 2 pixels vs. 3 pixels or more). For the case of a planar slope, the two measurements are of course equal. For the data presented in Figure 5, I suspect the difference would be imperceptible, and much smaller than issues related to DEM quality/resolution.

[revised manuscript text omitted]

**a.** Map view

[Figure]

**b.** Cross section in direction $\varphi$

[Figure]

**c.** Cross section (zoomed in)

[Figure]

Figure 1: **Model catchment setup, showing (a) map view, (b) cross section along azimuthal angle $\varphi$ (note that $|\gamma| \cancel{\gamma} = \alpha$ for $\varphi = 0$), and (c) close up of hillslope cross section.**

[Figure]

**Figure 2: Dipping catchment shielding geometry, illustrated using example from the San Gabriel Mountains, California, USA. Image is centered on 34.20°N, 117.61°W. Colored lines indicate planes fit through bounding ridgelines dipping at angle $\beta$. $\overline{S_0}$ indicates mean surface skyline shielding parameter calculated using algorithm of Codilean (2006), and $\overline{C}_{eff}$ indicates the mean total effective shielding factor calculated from the simplified catchment geometry.**

[Figure]

**Figure 3: Total shielding factor, $S(x, z)$, as a function of normalized vertical depth and distance from ridgeline for varying hillslope angle, $\alpha$.**

[Figure]

**Figure 4: Plots of (a) surface skyline shielding factor (b) normalized effective vertical attenuation length, and (c) total effective shielding factor as a function of distance from ridgeline for model runs with α = 0-80°. Dashed line in (a) indicates topographic shielding calculation using algorithm of Codilean (2006) applied to a digital elevation model of the case *α* = 80°.**

[Figure]

**Figure 5: Plot of normalized production rate relative to horizontal unshielded surface as a function of normalized vertical depth for a 60° slope with no additional skyline shielding. Near the surface, production rates are decreased due to slope shielding of incoming cosmic radiation; however, production rates at depth increase relative to the unshielded case due to additional radiation along shorter oblique pathways (Fig. 1c).**

[Figure]

**Figure 6: Plots showing mean total shielding factor, $\overline{C}_{eff}$, for (a) simple horizontal catchment case (Fig. 1) for varying mean hillslope angle, $\alpha$, which is equivalent to the "interior" dipping catchment case as a function of catchment dip, $\beta$ (Fig. 2), and (b), the mean total shielding factor, $\overline{C}_{eff}$, for the "exterior" dipping catchment model as a function of catchment dip, $\beta$ (Fig. 2). Red points in (a) indicate relationship between the mean surface skyline shielding factor, $\overline{S}_0$, as a function of mean hillslope angle for compilation of catchment [10]Be data in the Himalaya and Eastern Tibet as reported by Scherler et al. (2017). Red and yellow squares indicate mean surface skyline factor, $\overline{S}_0$, calculated for example catchments from San Gabriel Mountains (Fig. 2). Arrows indicate difference between mean surface skyline shielding factor and mean total shielding factor, $\overline{C}_{eff}$.**